# Zika, Flavivirus and Malaria Antibody Cocirculation in Nigeria

**DOI:** 10.3390/tropicalmed8030171

**Published:** 2023-03-14

**Authors:** Peter Asaga Mac, Axel Kroeger, Theo Daehne, Chukwuma Anyaike, Raman Velayudhan, Marcus Panning

**Affiliations:** 1Institute of Virologie Hermann Herder Strabe, Universitatsklinikum Freiburg, 79104 Freiburg, Germany; 2Centre for Medicine and Society, University of Freiburg, 79085 Freiburg, Germany; 3Federal Ministry of Health, Abuja 900001, Nigeria; 4World Health Organization, 1211 Geneva, Switzerland

**Keywords:** Zika, flavivirus, co-circulation, IgG, sero-epidemiology, malaria parasite

## Abstract

**Introduction**. Arboviruses and malaria pose a growing threat to public health, affecting not only the general population but also immunocompromised individuals and pregnant women. Individuals in vulnerable groups are at a higher risk of severe complications from the co-circulation and transmission of ZIKV, malaria, and FLAVI fever. In sub-Saharan countries, such as Nigeria, these mosquito-borne infections have clinical presentations that overlap with other diseases (dengue, West Nile virus, and Japanese encephalitis, chikungunya, and O’nyong o’nyong virus), making them a diagnostic challenge for clinicians in regions where they co-circulate. Vertical transmission can have a devastating impact on maternal health and fetal outcomes, including an increased risk of fetal loss and premature birth. Despite the global recognition of the burden of malaria and arboviruses, particularly ZIKV and other flaviviruses, there is limited data on their prevalence in Nigeria. In urban settings, where these diseases are endemic and share common biological, ecological, and economic factors, they may impact treatment outcomes and lead to epidemiological synergy. Hence, it is imperative to conduct sero-epidemiological and clinical studies to better understand the disease burden and hidden endemicity, thereby enabling improved prevention and clinical management. **Method**. Serum samples collected from outpatients between December 2020 and November 2021 in three regions of Nigeria were tested for the presence of IgG antibody seropositivity against ZIKV and FLAVI using immunoblot serological assay. **Results**. The overall cohort co-circulation antibody seropositivity of ZIKV, FLAVI and malaria was 24.0% (209/871). A total of 19.2% (167/871) of the study participants had ZIKV-seropositive antibodies and 6.2% (54/871) were FLAVI-seropositive, while 40.0% (348/871) of the subjects had malaria parasite antigens. Regional analysis revealed that participants from the southern region had the highest antibody seropositivity against ZIKV (21.7% (33/152)) and FLAVI (8.6% (13/152)), whereas those from the central region had a higher malaria parasite antigen (68.5% (287/419)). **Conclusions**. This study represents the largest comparative cross-sectional descriptive sero-epidemiological investigation of ZIKV-FLAVI and malaria cocirculation in Nigeria. The findings of this study revealed increased antibody seropositivity, hidden endemicity, and the burden of ZIKV, FLAVI, and malaria co-circulating in Nigeria.

## 1. Background

Arboviruses are one of the primary causes of mortality worldwide, posing a global public health challenge given their widespread diffusion and lack of broad-spectrum antivirals for prophylactic or therapeutic use [1]. Of particular concern is the Zika virus (ZIKV) and other flaviviruses (FLAVI), which are hyperendemic in numerous countries in the tropics and subtropics [1]. The first case of ZIKV infection was reported in Nigeria in 1975 [2]. These mosquito-borne infections produce clinical presentations that overlap with dengue, West Nile virus, and Japanese encephalitis, creating a diagnostic challenge for clinicians in regions where they co-circulate, including Nigeria [1]. As of December 2021, 89 countries and territories have confirmed evidence of autochthonous mosquito-borne transmission of the Zika virus (ZIKV) and other flaviviruses [3]. Evidence of ZIKV and other flavivirus transmission has been identified in several African countries; however, information on the current co-circulation incidence and trends of ZIKV, FLAVI, and malaria transmission remains limited [2,3].

The Zika virus (ZIKV) is primarily transmitted through the bite of infected *Aedes aegypti*, *Aedes albopictus*, *Aedes africanus*, and *Aedes hensilli*, which are some of the species that have been implicated in the transmission of Zika [4,5,6,7,8,9,10]. Other modes of ZIKV and flavivirus transmission include sexual, perinatal, and congenital transmission. Zika infection is also associated with various pregnancy-related complications, including preterm birth and miscarriage. There is an increased risk of neurological sequelae, including Guillain-Barré syndrome, neuropathy, and myelitis in both adults and children. Neonates exposed to ZIKV infection during gestation have elevated immunoglobulin levels in the CSF, particularly those who develop microcephaly [11].

The emergence of the Zika virus (ZIKV) and other flaviviruses in malaria-endemic regions has created intriguing but potentially alarming scenarios [5]. It has been postulated by a study conducted in Sri Lanka that for Zika and dengue viruses, prior infection with one virus modulates the severity of subsequent infection with the other virus [6,7,12,13,14,15]. This poses a major health concern, because of the high homology between these arboviruses, cross-reactivity of antibodies against flaviviruses such as DENV, yellow fever virus (YFV), tick-borne encephalitis virus (TBEV), and Japanese encephalitis virus (JEV) can occur, which may complicate the interpretation of serological results, cocirculation, and severe fetal outcomes. The actual burden in the epidemic and interepidemic periods is grossly unknown and underreported in Nigeria [8]. Vector-borne diseases such as Malaria, FLAVI, and ZIKV have similar endemic profiles and symptoms, with potentially fatal outcomes if left undetected. It is crucial to comprehend the prevalence and distribution of these diseases to enhance diagnosis and develop therapeutic interventions. These infections have recently been a worldwide concern, particularly in tropical and subtropical areas like Nigeria, due to recurring outbreaks. Several of these diseases have become endemic to regions where malaria is widespread, causing millions of occurrences each year. Malaria is a significant public health threat and is caused and spread by five different species of the protozoal parasite, *Plasmodium.* These include *P. falciparum*, *P. ovale*, *P. malariae*, *P. vivax*, and *P. knowlesi*, which are carried and spread by *Anopheles* mosquito [4,5]. Malaria, ZIKV, and FLAVI have a similar epidemic pattern that mostly affects tropical regions worldwide. Several studies have shown that all three diseases can co-circulate [10,16]. The diseases share comparable clinical manifestations where fever is the most common symptom. The burden of these infections has surged due to frequent outbreaks in different regions of Nigeria. Factors such as global travel and rapid urbanization have introduced vector populations to new environments, contributing to the expansion of disease endemicity [12]. Consequently, the co-circulation of Malaria with FLAVI and ZIKV complicates the diagnosis and treatment process.

Despite this trend and the potential public health threat, there is no reliable data, and little is known about the co-circulation of ZIKV, malaria, and other flavivirus infections in Nigeria. We investigated the seroprevalence of ZIKV malaria and FLAVI and their possible cocirculation (participants who were serologically positive for ZIKV, malaria, and other flaviviruses during the sampling period or time) in three regions of Nigeria [13,14,15].

## 2. Methods

### 2.1. Study Design and Site

A cross-sectional study was conducted at three university teaching hospital centers in Nigeria: namely, the Federal Medical Centre, Keffi, located in Nasarawa State; the Central Nigeria Abia State University Teaching Hospital, Aba, located in Abia State, Southern Nigeria; and the Baru-Diko Teaching Hospital, Kaduna, Kaduna State, located in Northern Nigeria (Figure 1).

The three states have a population of over 30 million inhabitants. Forty-five percent of the population live in urban areas (urban settlement in the context of the present study refers to high human population density and infrastructure of the built environment), 40% live in rural areas (open countryside with population densities of less than 500 people per square mile or places with fewer than 1500 people), and 15% live in slums or informal settlements (informal settlements within urban cities with inadequate housing, squalids, and miserable living conditions) [1]. The average annual temperature ranges from 21 °C to 37 °C, whereas in the interior lowlands, temperatures are generally above 27 °C. The mean annual precipitation is 1165.0 mm. It rains throughout southern Nigeria but much less so in the central and northern regions, with episodes of flooding and other environmental catastrophes. Most of the rainfall occurs between April and October, with minimal rainfall occurring between November and March. The main occupation of the inhabitants of the three regions is farming at both the commercial and subsistence levels.

### 2.2. Study Population

The study population comprised outpatients, including pregnant women enrolled for antenatal care and patients presenting with illness, at the rapid-access healthcare and antiretroviral (people living with AIDS) units of the hospitals between December 2020 and November 2021. These hospitals were purposefully selected to reflect the diversity of different cultures, religions, ethnicities, topographical and vegetation features, and human activities in the three geographical regions. The inclusion criteria were all outpatients within an age range of 0 months to 80 years who agreed to participate in the study and signed the consent form, including children, whose parents or guardians gave consent. The exclusion criteria were participants who were already undergoing treatment, those who refused to sign the consent form, and seriously ill hospitalized patients.

### 2.3. Screening of the Study Participants

A standardized questionnaire containing questions on demographics, medical history, vital signs and symptoms, clinical evaluation, hospitalization data, and a summary form was used to collect the information. All the research participants were examined for malaria-, ZIKV-, and FLAVI-related symptoms (fever, headaches, rashes, joint pain, conjunctivitis, and muscle discomfort) (Table 1). Before enrolment, participants were provided with protocol-specific information and had this clearly explained to them in English and their respective native languages. After enrolment, the participants signed an informed consent form. The participants who were unable to read or write were asked to verbally assent and give their thumbprint to indicate their willingness to participate.

### 2.4. Total Number of Samples Collected

The simple random sampling method was used to collect 871 samples from participants in the three regions. In total, 262 samples were collected from outpatients, 499 from HIV-positive patients, and 110 from blood banks. The sample size calculation (based on a 40% expected proportion of ZIKV and FLAVI in a total population of 500,000 patients with a 95% confidence interval and a *p*-value of 0.05) [17] indicated a minimum sample size of 384 serum samples, which was increased to 871 samples for subgroup analyses by region.

### 2.5. Laboratory Testing

All study participants provided 5 mL of venous blood, and a local clinical diagnostic laboratory technician collected the blood samples from the three blood banks. All participants were screened for malaria using an RDT specific for the parasite (SD BIOLINE Malaria Differential P.f/Pan Ag RDT (HRP II+ pLDH, Abbott, Mikrogen Diagnostik, Neuried, Germany). In brief, 5 µL of blood sample was transferred into the sample well using the appropriate device included in the kit, and five drops of lysis buffer were added to the buffer well. The results were read visually after 15–20 min. Screened samples were shipped on dry ice to the Institute of Virology Universitatsklinikum, Freiburg, where they were tested for the presence of human immunoglobulin G (IgG) antibodies using recomLine Tropical Fever for the presence of arboviral antibody serological marker IgG immunoblot (Mikrogen Diagnostik, Neuried, Germany) ZIKV NS1 and ZIKV Equad and flavivirus according to the manufacturer [18]. In brief, test strips were loaded with ZIKV and FLAVI antigens and incubated with diluted serum in a dish for 1 h. The strips were then washed three times. Peroxidase-conjugated anti-human antibodies (IgG-specific) were added, incubated for 45 min, and washed three times. Insoluble bands developed at the sites on the test strips occupied by antibodies 8 min after the addition of the coloring solution.

### 2.6. Statistical Tests

Statistical analyses were performed using SPSS version 21. Descriptive statistics were employed for the analysis of the results, and 95% confidence intervals [CI] were used to identify the sociodemographic and behavioral characteristics of the study population. The results are presented in tables and figures. The chi-squared test was performed and deemed statistically significant at *p* ≤ 0.05.

## 3. Ethics Statement

The study protocol was reviewed and approved by the local ethics committee on human research at the University of Freiburg [No. 140/19], and National Ethics Committee on Human Research of Nigeria [No. KF/REC/02/21].

## 4. Results

### 4.1. Signs and Symptoms Presented by ZIKV, FLAVI Malaria Monoinfected Patients

The participants in this study who were seropositive for flaviviruses (arboviral infections) and malaria presented with undifferentiated clinical signs and symptoms associated with acute febrile illness (AFI) during the study. Table 1 shows that the only difference between flavivirus- and ZIKV-seropositive participants was the extent of clinical manifestation, which was more severe in FLAVI (50.0%) and ZIKV (25.0%) seropositive participants than in malaria (4.7%) seropositive participants. (Table 1).

### 4.2. Arbovirus Serology

The presence of mild and non-specific symptoms necessitated the use of serological tests in epidemiological studies. Cross-reactive antibodies between flaviviruses and alphaviruses may complicate the interpretation of serological tests (immunoblot assays). Consequently, ZIKV and FLAVI seropositive results were interpreted as flavivirus-seropositive. This study had several limitations. The cross-reactivity of IgG antibodies between flaviviruses and other alphaviruses is well-established and a confounding factor in serological studies investigating the seropositivity of arboviruses. Because of the large sample size, it was impractical to conduct additional testing using techniques such as the plaque reduction neutralization test (PRNT), other seroneutralization tests, and PCR.

### 4.3. Demographic Characteristics of Participants and ZIKV, FLAVI, and Malaria Antibody Seropositivity

A total of 871 participants were recruited from three geographical regions in Nigeria. Among them, 17.5% (152/871) were from Abia (Southern Nigeria), 34.4% (300/871) were from Kaduna (Northern Nigeria), and 48.1% (419/871) were from Nasarawa (Central Nigeria). Of the 871 subjects, 71.0% (619/871) were female and 29.0% (252/871) were male. Of these, 57.3% (499/871) were HIV-positive, 2.7% (233/871) were pregnant, 58.2% (507) were from urban areas, 29.6% (258/871) were from rural areas, and 12.2% (106/871) were from slums or informal settlements. Of the 871 serum samples, 87.4% (761/871) were from outpatients, and 12.6% (110/871) were from blood banks. The age of the participants ranged from 0 months to 80 years, with a mean age of 36.6 years.

The overall cohort co-circulation antibody seropositivity of ZIKV, FLAVI, and malaria was 24.0% (209/871). A total of 19.2% (167/871) of the study participants had ZIKV-seropositive antibodies and 6.2% (54/871) were FLAVI-seropositive, while 40.0% (348/871) of the subjects had malaria parasite antigens.

Regional analysis revealed that participants from the southern region had the highest antibody seropositivity against ZIKV (21.7% (33/152)) and FLAVI (8.6% (13/152)), whereas those from the central region had a higher malaria parasite antigen (68.5% (287/419)). However, the odds of FLAVI were 1.5 times more in the central region than in the other two regions. (Table 2).

### 4.4. Sex-Specific Antibody Seropositivity of Arboviral Infection

Male participants had slightly higher antibody seropositivity against ZIKV (20.6% (52/252)), whereas female participants had marked antibodies against FLAVI (6.8% (42/619)) and malaria antigen (62.7% (388/619)). (Table 2).

### 4.5. Place-Specific Antibody Seropositivity of Arboviral Infection

In the present study, ZIKV seropositive antibodies were slightly more evident in slums (19.8% (21/106)) and rural participants (19.8% (51/258)) than in urban participants. (Table 2). FLAVI-seropositive antibodies (10.4% (11/106)) and malaria parasite antigen (97.2% (103/106)) were more prevalent in slums. The odds of malaria were 22.9 times more in the slum group than in the other groups (*p* = 0.00). (Table 2)

### 4.6. ZIKV, FLAVI, and Malaria Antibody Seropositivity in Pregnant and Non-Pregnant Participants

Antibody seropositivity against ZIKV, (18.8% (125/638)), FLAVI (6.4% (41/638)), and malaria (47.6% (304/638)) was slightly more prevalent and elevated among non-pregnant subjects than in pregnant subjects. (Table 2).

### 4.7. HIV Status-Specific Antibody Seropositivity

The HIV-negative group had the highest levels of detectable antibodies against ZIKV 24.4% (70/312) and malaria antigen (53.8% (200/372)), whereas FLAVI-seropositive antibody was more prevalent in the HIV-positive group 35 (7.0% (35/499)) (*p* = 0.00). (Table 2).

### 4.8. Blood Product Antibody Seropositivity

Sera from the blood banks showed remarkable antibodies against ZIKV 62.7% (69/110) and FLAVI 38.2% (42/110), whereas significantly lower antibody seropositivity was observed from outpatients’ serum samples. Interestingly, malaria parasite antigens were more evident in the outpatient sera (*p* = 0.00) (Figure 2 and Table 2).

### 4.9. Age-Specific Antibody Seropositivity of ZIKV, FLAVI, and Malaria Co-Circulation

The highest detectable seropositive antibody against ZIKV was obtained in the 50–59-year-old (32.9% (27/82)) and 70–79-year-old (25.0% (2/8)) age groups. However, a much lower seroprevalence was observed in the 0–9-year-old age group. Similarly, seropositive antibodies against FLAVI were highest in the 80-year-old (16.7% (1/6)) and 50–59-year-old (8.5% (7/82)) age groups, while the malaria parasite seropositive antigen was considerably higher in the 20–29-year-old (82.1% (161/196)), 10–19-year-old (72.5% (37/51)) and 0–9-year-old (66.7% (2/3)) age groups, (*p* = 0.00) (Table 3).

### 4.10. Sociodemographic Characteristics and Cocirculation Antibody Seropositivity of ZIKV, FLAVI and Malaria

The overall cohort antibody seropositivity against ZIKV-FLAVI was 17.5% (153/871), whereas that against FLAVI-malaria was 2.2% (19/871) and ZIKV-malaria was 4.3% (37/871) (Table 4). The regional subgroup analysis revealed the highest ZIKV-FLAVI (7.2% (11/152)), FLAVI-malaria (3.9% (6/152)), and ZIKV-malaria (7.2% (11/152)) cocirculation antibody seropositivity was in the southern region, whereas much lower antibody seropositivity was observed in the central and northern regions. However, the odds of ZIKV-FLAVI and ZIKV-malaria cocirculation seropositive antibodies were 2.8 times more in the northern region than in all the other regions (Figure 3 and Table 4).

### 4.11. Sex-Specific Cocirculation Antibody Seropositivity against ZIKV, FLAVI and Malaria

Sex-specific cocirculating antibodies against ZIKV-FLAVI (3.6% (9/252)), FLAVI-malaria (2.4% (6/252)), and ZIKV-malaria (3.6% (9/252)) were observed to be more prevalent in male participants than in female participants. However, the odds of co-circulating antibody seropositivity for both infections were higher in the female group than in the male group (Table 4).

### 4.12. Place-Specific Cocirculation Antibody Seropositivity against ZIKV, FLAVI and Malaria

The seroprevalence for ZIKV-FLAVI (13.2% (14/106)), FLAVI-malaria (11.3% (12/106)), and ZIKV-malaria (13.2% (14/106)) was the highest in the slum group, whereas a much lower seroprevalence was observed in the rural and urban groups. The odds of antibody seropositivity cocirculation were greater in the slum group than in the rural and urban groups (*p* = 0.00) (Table 4).

### 4.13. Pregnancy Status-Specific Cocirculation Antibody Seropositivity against ZIKV, FLAVI and Malaria

The cocirculating antibody seropositivity was more remarkable in pregnant participants [(ZIKV-FLAVI (6.9% (16/233)), FLAVI-malaria (31.3% (3/233), and ZIKV-malaria (6.9% (16/233) than in non-pregnant participants. The odds of co-circulating antibody seropositivity were higher in the non-pregnant group than in the pregnant group (Table 4).

### 4.14. HIV-Status-Specific Cocirculation Antibody Seropositivity against ZIKV, FLAVI and Malaria

HIV-positive patients had the most ZIKV-FLAVI (3.0% (15/499) and ZIKV-malaria (3.0% (15/499) cocirculating antibodies, whereas the most seropositive cocirculating antibodies against FLAVI-malaria were observed in the HIV-negative group (3.0% (11/372). The odds of ZIKV-FLAVI and ZIKV-malaria cocirculating antibodies were higher in the HIV-negative group than in the HIV-positive group (Table 4).

### 4.15. Blood Product-Specific Cocirculation Antibody Seropositivity against ZIKV, FLAVI and Malaria

Serum samples from the blood banks showed the highest cocirculating antibody seropositivity against ZIKV-FLAVI (13.6% (15/110) and ZIKV-malaria (13.6% (15/110) compared to sera from outpatient participants, while seropositive antibodies against FLAVI-malaria (1.6% (12/761)) were more evident in serum samples from outpatient participants (Table 4).

### 4.16. Age-Specific Cocirculation Antibody Seropositivity against ZIKV, FLAVI and Malaria

The overall age cohort of ZIKV-FLAVI, FLAVI-malaria, and ZIKV-malaria-cocirculating antibodies in the study population was 10.5% (89/871). The age subgroup revealed the highest cocirculating antibody seropositivity against ZIKV-FLAVI in the 70–79-year-old (25.0% (2/8)) and 80-year-old (16.7% (1/6)) age groups, whereas the 60–69-year-old age group (3.7% (1/27)) had the highest cocirculating antibody seropositivity against FLAV-malaria. ZIKV-malaria-seropositive co-circulating antibodies were more prevalent in the 10–19-year-old (9.8% (5/51)) and 60–69-year-old (7.4% (2/27)) age groups (Table 5).

### 4.17. Monthly Antibody Seropositivity ZIKV, FLAVI and Malaria during the Sampling Period

An assessment of the monthly trends for the two arboviruses and malaria revealed a significant antibody seropositivity distribution throughout the sampling period. ZIKV antibody seropositivity was particularly high during the two years of the sampling period compared with FLAVI-seropositive antibodies and malaria (Figure 4).

## 5. Discussion

Our study is one of the few published studies on the seroprevalence, burden, hidden endemicity, and geographic spread of arboviral and malaria infections in sub-Saharan Africa. The results indicated that Zika, flaviviruses, and malaria cocirculation antibodies have spread across the three Nigeria regions, with a seroprevalence of 24.0%. The study revealed that 19.2% of the study population had ZIKV seropositive antibodies, 6.2% had FLAVI seropositive antibodies, and 40.0% had malaria seropositive antigens. It was also observed that 7.5% had ZIKV-FLAVI cocirculation antibody seropositivity, 4.3% had ZIKV-Malaria cocirculation seropositive antibodies, and 2.2% had FLAVI-malaria cocirculation seropositive antibodies. In several studies, it has been suggested that antibody seropositivity against ZIKV is notably lower in southern Nigeria. These studies include those conducted by Oyefolu et al. (2.0%) [13], Otu et al. in eastern Nigeria (12.0%) [4], Mathe et al. (9.0%) [7] and Anejo et al. (14.4%) [8]. However, Fagbemi et al. reported a much higher antibody seropositivity rate (31.0%) in southern Nigeria.

Seroprevalence rates of 0.7% and 27.3% of arboviral infections have been recorded in Kenya and Ethiopia, respectively [19,20]. There is no known definitive signature work on the cocirculation of malaria, ZIKV, and FLAVI cocirculation antibodies in Nigeria.

This study also explains the recent increase in the antibody seroprevalence of ZIKV, FLAVI, and malaria in Nigeria. The reasons for the increase in the antibody seropositivity of ZIKV, FLAVI, and malaria could not be determined in the current study but could be attributed to antibody cross-reactivity with other arboviruses [15], arboviral vaccine (particularly yellow fever vaccine) urbanization, high human population density, unplanned settlements, poor drainage systems in major cities in the three regions, inadequate and unwholesome waste disposal, stagnant water bodies, and water collected in waste metal containers and vehicle tires. These microhabitats serve as breeding sites for *Aedes* spp. mosquitoes, which are vectors of arboviral transmission [7,8,14,15]. The antibody seropositivity of these two arboviruses and malaria parasite cocirculations are also shaped or impacted by numerous factors that affect various natural elements, such as the survival, reproduction, development, activity, distribution, and abundance of vectors and hosts. Additionally, these factors impact pathogen development, maintenance, replication, and transmission. The range of pathogens, vectors, and hosts is also influenced by these natural factors, along with human behavior, which can also influence disease outbreak frequency, onset, and distribution. While the route of transmission is typically identifiable, distribution within an endemic area can often be uneven. Malnourished individuals and those with weakened immune systems are particularly vulnerable to these types of diseases. Other studies report similar and contrasting findings [1,2,12,13,17,18,19,20,21]. These results could also have been influenced by the sampling period.

IgG antibodies against ZIKV and FLAVI were more prevalent in the southern region than in the northern and central regions, which could be attributed to human mobility and the displacement of large populations and communities due to political fatigue or communal clashes in southern Nigeria [8,12,14]. Massive population movements in urban cities in search of better working conditions have also contributed to changes in the seropositivity of the two arboviral diseases and malaria, thereby altering and complicating their diagnosis and treatment outcomes [8,9,10,16,22]. The frequency of extreme climate events, such as floods, vegetation index, and widespread changes in mosquito habitats and ecosystems in the central and southern regions may have also shaped seropositivity in various regions and participants. Similar findings have been reported in various studies conducted in Nigeria [2,7,8,12,14].

Sociodemographic factors, such as age, sex, place of domicile, pregnancy status, and malaria cocirculation in the population, also revealed a high rate of antibody seropositivity, hidden endemicity, and heavy and unrecognized burdens of ZIKV, FLAVI, and malaria in the three regions. This may be explained by the silent cocirculation of these arbovirus vectors within the three geographical regions, as well as different socioeconomic factors such as poor housing, cultural norms and behaviors, level of education, mode of farming, and other agricultural and commercial activities in the three regions. The different antibody seropositivity levels across the three regions could potentially be related to the overlapping or widespread distribution of *Aedes* and *Anopheles* spp. mosquito vectors in all three regions [1,2,3,4]. Variations in climate (increasing reproductive activities and shortening the extrinsic cycle of CHIKV and DENV in the vector) could be related to different types of vegetation (from dense in southern Nigeria to savanna grassland in the central region and arid vegetation in the north), disparate meteorological factors, vector reproductive indices, attack rates of the vectors, sampled population, molecular diagnostic tools, or assays applied [14,15,17,21]. The discrepancies in antibody seropositivity could also be attributed to poor health facilities in the three regions, underestimated or undetected arboviral infections by various laboratories, limited facilities, inadequate testing capacity, and trained manpower. This is the exact opposite of the situation in developed countries, where most cases of arboviral infections imported from developing countries are readily detected [1,2,3,4,12,13,14,15,22,23].

ZIKV and FLAVI IgG seroprevalence was particularly higher among adults 20–80 years of age compared to those aged <20 years, which is consistent with the increased vector exposure related to socioeconomic activities among various demographic groups [1,2,3,4,5,6,7,8,9,10]. Several factors contribute to this, including past infections (exposure over time), imunosenescence in old age, and long-standing immunity to arboviruses and malaria [1,12,14]. Older people are also more likely to be bitten by *Aedes* mosquitoes because they sit in unscreened areas for long periods (daytime feeding activities of *Aedes aegypti*). Maternal immunity might have contributed to the low seropositivity in children because of the increase in IgG levels, which acts as a protective factor in infants. However, an increase in malaria-seropositive antigens was observed in the younger age group. This could be attributed to the waning of immunity acquired during birth [7,8,14].

Seropositive anti-ZIKV, FLAVI, and malaria antigens were markedly detected in slum participants. This phenomenon may be attributed to several factors, including rural-urban migration resulting from political conflict or exhaustion, especially in northern and central Nigeria, and travel and commercial activities that lead to overcrowding, and poor refuse disposal dumpsites, unhygienic sewage and drainage systems, stagnant water in tires, and tin containers, which serve as suitable habitats for Aedes species [1,4,5,6,7,8,13], resulting in outbreaks of unknown ZIKV and FLAVI in these three regions [5,6,7,8,13,14,21]; the expansion of agricultural activities into sylvatic rural areas could also shape the transmission dynamics of these diseases [1]. Antibody seropositivity for ZIKV, FLAVI, and malaria was evident in HIV-positive and HIV-negative individuals. These findings seem ambiguous, but the most plausible explanation could be that most HIV-positive study participants adhered to antiretroviral medication [9] because most study participants were recruited from ART units in the hospitals. According to several studies, HIV-positive individuals receiving ART may not be at a higher risk of malaria and arboviral co-circulation complications than the HIV-negative population or the general public. It is also uncertain whether short- or long-term repercussions may arise from malaria, ZIKV, and FLAVI co-circulation in HIV-positive individuals [19].

Seropositivity was particularly higher in non-pregnant women than in pregnant women. This finding contrasts with several other studies that have reported a higher prevalence [16,22,23] in pregnant women. The differences in the current study could be attributed to the health-seeking behaviors of pregnant women, as most pregnant women were recruited from the antenatal units of tertiary hospitals, resulting in strict adherence to antimalarial medication and the maintenance of high health standards during pregnancy.

There were marked ZIKV and FLAVI seropositive antibodies with considerable malaria-parasite antigen among the participants, and it remains imprecise whether the presence of co-circulating seropositive antibodies increases or reactivates malaria or vice versa.

Several studies have found that simultaneous or co-circulation of malaria and arboviruses may increase seroprevalence rates, especially in tropical and subtropical areas [14,15,21]. It also indicates silent transmission of arboviruses within various communities in the three regions [14,21].

Serum samples from the blood banks showed the most seropositive antibodies against ZIKV and FLAVI mono- and co-circulating antibodies. This could be attributed to blood donated by asymptomatic individuals and the failure, lack, and inability of health services to diagnose and distinguish between malaria; they typically screen for malaria, but not arboviral infections or other febrile illnesses.

## 6. Limitations

The current study was a sentinel serosurvey in tertiary hospitals in three regions of Nigeria; therefore, it may not be representative of true Zika and flavivirus seropositivity. Additionally, we did not perform plaque reduction neutralization tests (PRNT) or PCR to confirm the presence of flavivirus and ZIKV, and there was a likelihood of false negatives or positives due to cross-reactivity and arboviral vaccines. There were more females than males, and different age groups, which may have led to bias and confounding variables.

## 7. Conclusions

Although the global burden of malaria and arboviruses (ZIKV and FLAVI) is well recognized, there is a dearth of knowledge and data about malaria, ZIKV, and FLAVI, especially in Nigeria. It is also intriguing and unknown whether these vector-borne infections are transmitted by a single bite of a mosquito, or whether during the feeding process, a single mosquito could pick up both infections. This study represents the largest comparative cross-sectional descriptive sero-epidemiological investigation of ZIKV-FLAVI and malaria cocirculation in Nigeria. Findings from this study revealed increased antibody seropositivity, hidden endemicity, and the burden of ZIKV, FLAVI, and malaria co-circulating in Nigeria. These results may have been shaped by antibody cross-reactivity, past exposure to the arboviral vaccine (possibly the yellow fever vaccine), long-term exposure immunity, different vector densities due to different vegetation types across the three regions, human population indices or anthropogenic activities, climate change, vector adaptations, variations in temperature and humidity, and flooding, which may have shaped the habitats and microclimates of the three regions.

Therefore, differential diagnosis should be performed in patients with acute febrile syndrome, and screening of blood donors for arboviral infections will assist clinicians and policymakers in designing interventions, generating data, and implementing effective control measures.

## Figures and Tables

**Figure 1 tropicalmed-08-00171-f001:**
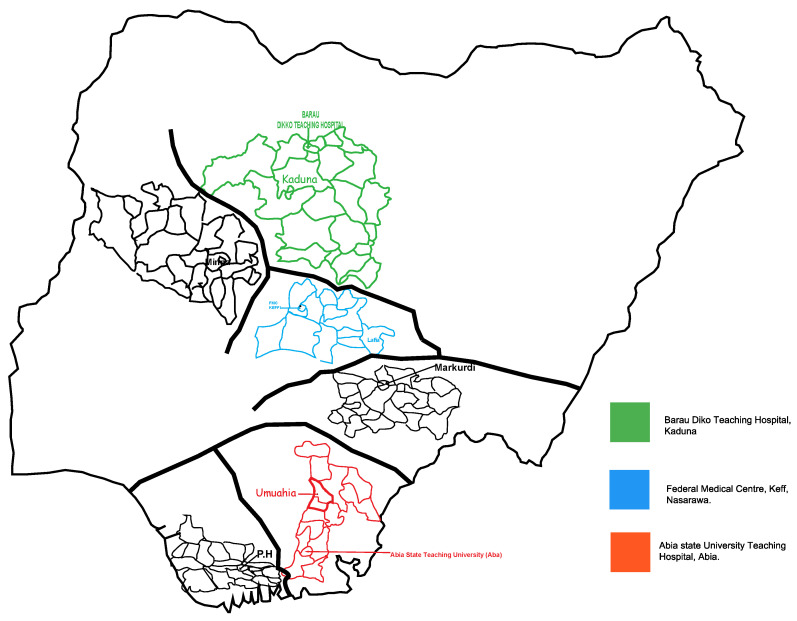
Arboviral and malaria study sites in Nigeria.

**Figure 2 tropicalmed-08-00171-f002:**
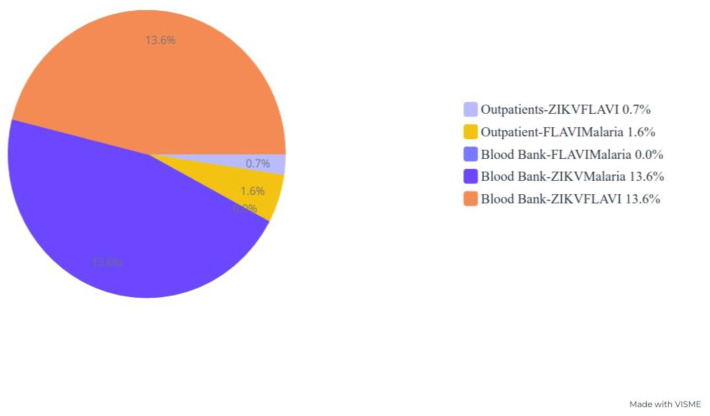
Blood product antibody seropositivity.

**Figure 3 tropicalmed-08-00171-f003:**
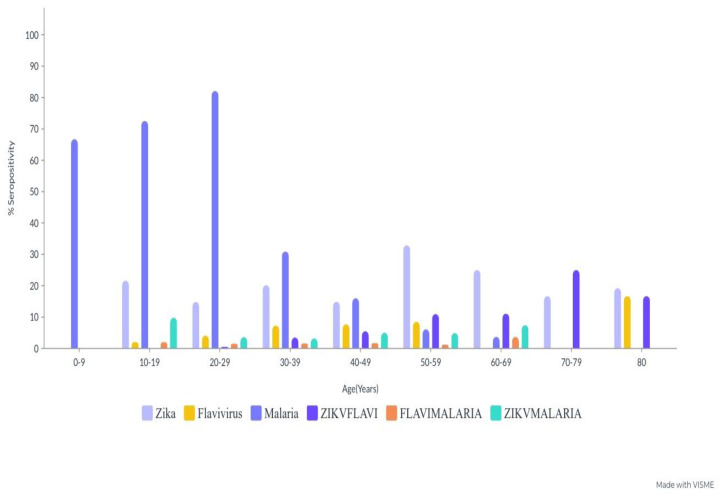
Age-specific antibody seropositivity of ZIKV, FLAVI, and malaria cocirculation.

**Figure 4 tropicalmed-08-00171-f004:**
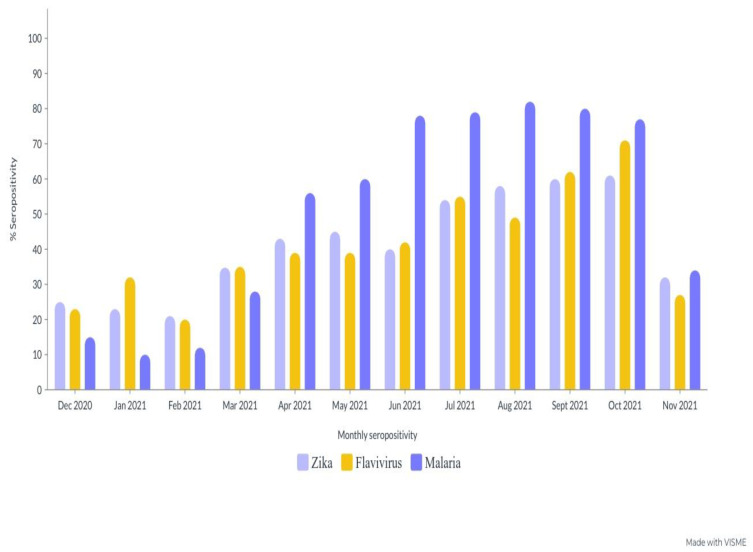
Monthly antibody seropositivity for ZIKV, FLAVI, and malaria during the sampling period.

**Table 1 tropicalmed-08-00171-t001:** Signs and symptoms presented by ZIKV, FLAVI, and malaria monoinfected patients.

Sign and Symptoms	Mono-Infection (% Sign & Symptoms)		
	Anti-Zika Positive (N = 167)	Anti-Flavivirus Positive (N = 54)	Anti-Malaria Positive (N = 362)
Headaches	43.7% (73/167)	79.6% (43/54)	83.4% (302/362)
Exanthema	33.5% (56/167)	18.5% (10/54)	5.5% (20/362)
Fever	61.1% (102/167)	72.2% (39/54)	88.4% (320/362)
Abdominal pain	21.0% (35/167)	62.9% (34/54)	55.5% (201/362)
Diarrhoea	13.8% (23/167)	7.4% (4/54)	2.8% (10/362)
Myalgia	38.3% (64/167)	74.1% (40/54)	27.1% (98/362)
Vomiting	32.9% (55/167)	22.2% (12/54)	18.5% (67/362)
Generalized body pains	83.8% (140/167)	72.2% (39/54)	58.6% (212/362)
Arthralgia	36.5% (61/167)	38.9% (21/54)	21.5% (78/362)
Edema	1.2% (2/167)	12.9% (7/54)	3.8% (14/362)
Hemorrhagic manifestation	0.0% (0/167)	3.7% (2/54)	2.5% (9/362)
Retro-orbital pain	12.6% (21/167)	24.1% (13/54)	16.3% (59/362)
Nausea	7.2% (12/167)	59.3% (32/54)	13.5% (49/362)
Non-purulent conjunctivitis	19.2% (32/167)	22.2% (12/54)	0.6% (2/362)
Leukopenia	25.7% (43/167)	50.0% (27/54)	4.7% (17/362)

**Table 2 tropicalmed-08-00171-t002:** Demographic characteristics of participants and ZIKV, FLAVI, and malaria antibody seropositivity.

Region	Zika Virus (ZIKV)						Flavivirus (FLAVI)							Malaria					
	Negative	Negative	Positive	Total Examined (N)	95% CI	OR	*p-Value*	Negative	Positive	Total Examined (N)	95% CI	OR	*p-Value*	Negative	Positive	Total Examined (N)	95% CI	OR	*p-Value*
South	79 (52.0%)	119 (78.3%)	33 (21.7%)	152 (100%)		1	0.85	139 (91.4%)	13 (8.6%)	152 (100%)		1	0.18	50 (32.9%)	102 (67.1%)	152 (100%)		1	0.67
North	99 (33.0%)	245 (81.7%)	55 (18.3%)	300 (100%)	0.65–1.41	1.0		287 (95.7%)	13 (4.3%)	300 (100%)	0.32–1.24	0.6		99 (33.0%)	201 (67.0%)	300 (100%)	0.68–1.28	0.9	
Central	128 (30.5%)	340 (81.1%)	79 (18.9%)	419 (100%)	0.70–1.51	1.0		391 (93.3%)	28 (6.7%)	419 (100%)	0.80–3.10	1.5		132 (31.5%)	287 (68.5%)	419 (100%)	0.78–1.47	1.0	
Sex																			
Male	90 (35.7%)	200 (79.4%)	52 (20.6%)	252 (100%)		1	0.48	240 (95.2%)	12 (4.8%)	252 (100%)		1	0.00	147 (58.3%)	105 (41.7%)	252 (100%)		1	0.00
Female	216 (34.9%)	504 (81.4%)	115 (18.6%)	619 (100%)	0.79–1.64	1.1		577 (93.2%)	42 (6.8%)	619 (100%)	0.35–1.32	0.6		231 (37.3%)	388 (62.7%)	619 (100%)	0.31–0.57	0.4	
Domicile																			
Urban	197 (38.9%)	412 (81.3%)	95 (18.7%)	507 (100%)		1	0.99	483 (95.3%)	24 (4.7%)	507 (100%)		1	0.26	60 (23.3%)	198 (76.7%)	258 (100%)		1	0.00
Rural	78 (30.2%)	207 (80.2%)	51 (19.8%)	258 (100%)	0.56–1.75	1.0		239 (92.6%)	19 (7.4%)	258 (100%)	0.57–2.79	1.3		203 (40.1%)	304 (59.9%)	507 (100%)	0.01–0.13	0.4	
Slum	31 (29.2%)	85 (80.2%)	21 (19.8%)	106 (100%)	0.56–1.76	1.0		95 (89.6%)	11 (10.4%)	106 (100%)	0.35–1.72	0.8		3 (2.8%)	103 (97.2%)	106 (100%)	7.17–73.2	22.9	
Pregnancy status																			
Pregnant	88 (37.8%)	191 (82.0%)	42 (18.0%)	233 (100%)		1	0.60	220 (94.4%)	13 (5.6%)	233 (100%)		1	0.64	135 (57.9%)	98 (42.1%)	233 (100%)		1	0.14
Nonpregnant	218 (34.2%)	513 (81.2%)	125 (18.8%)	638 (100%)	0.61–1.32	0.9		597 (93.6%)	41 (6.4%)	638 (100%)	0.45–1.63	0.8		334 (52.4%)	304 (47.6%)	638 (100%)	0.58–1.07	0.8	
HIV status																			
HIV positive	282 (56.5%)	429 (86.0%)	70 (14.0%)	499 (100%)		1	0.00	464 (93.0%)	35 (7.0%)	499 (100%)		1	0.00	376 (75.4%)	123 (24.6%)	499 (100%)		1	0.00
HIV negative	324 (87.1%)	275 (73.9%)	97 (26.1%)	372 (100%)	0.32–0.65	0.5		369 (99.2%)	3 (0.8%)	372 (100%)	2.83–30.4	9.2		172 (46.2%)	200 (53.8%)	372 (100%)	0.21–0.37	0.3	
Blood products																			
Outpatient serum	287 (37.7%)	663 (87.1%)	98 (13.9%)	761 (100%)		1	0.00	606 (79.6%)	155 (20.4%)	761 (100%)		1	0.00	372 (48.9%)	389 (51.1%)	761 (100%)		1	0.00
Blood bank serum	19 (17.3%)	41 (37.3%)	69 (62.7%)	110 (100%)	0.05–0.13	0.1		68 (61.8%)	42 (38.2%)	110 (100%)	0.27–0.63	0.4		110 (100%)	0 (0.0%)	110 (100%)	14.3–373	231	
Grand Total (N)	306 (35.1%)	704 (80.8%)	167 (19.2%)	871 (100%)	0.17–0.22			817 (93.8%)	54 (6.2%)	871 (100%)	0.5–0.7			523 (60.0%)	348 (40.0%)	871 (100%)	0.39–0.41		

**Table 3 tropicalmed-08-00171-t003:** Age-specific antibody seropositivity of Zika, FLAVI, and malaria cocirculation.

Age (Years)	Zika Virus					Malaria					Flavivirus				
	Negative	Positives	Total Examined (N)	95% CI	*p-Value*	Negative	Positive	Total Examined (N)	95% CI	*p-Value*	Negative	Positive	Total Examined (N)	95% CI	*p-Value*
0–9	3 (100%)	0 (0.0%)	3 (100%)	0	0.00	1 (33.3%)	2 (66.7%)	3 (100%)	0.65–0.67	0.03	3 (100%)	0 (0.0%)	3 (100%)	0	0.05
10–19	40 (78.4%)	11 (21.6%)	51 (100%)	0.20–0.22		14 (27.5%)	37 (72.5%)	51 (100%)	0.71–0.73		50 (98.0%)	1 (2.0%)	51 (100%)	0.1–0.3	
20–29	167 (85.2%)	29 (14.8%)	196 (100%)	0.13–0.15		35 (17.9%)	161 (82.1%)	196 (100%)	0.81–0.83		188 (95.9%)	8 (4.1%)	196 (100%)	0.3–0.5	
30–39	253 (79.8%)	64 (20.2%)	317 (100%)	0.19–0.21		219 (69.1%)	98 (30.9%)	317 (100%)	0.29–0.31		294 (92.7%)	23 (7.3%)	317 (100%)	0.6–0.8	
40–49	154 (85.1%)	27 (14.9%)	181 (100%)	0.13–0.15		152 (84.0%)	29 (16.0%)	181 (100%)	0.15–0.17		167 (92.3%)	14 (7.7%)	181 (100%)	0.6–0.8	
50–59	55 (67.1%)	27 (32.9%)	82 (100%)	0.31–0.33		77 (93.9%)	5 (6.1%)	82 (100%) 0	0.5–0.7		75 (91.5%)	7 (8.5%)	82 (100%)	0.7–0.9	
60–69	21 (77.8%)	6 (22.2%)	27 (100%)	0.21–0.23		26 (96.3%)	1 (3.7%)	27 (100%)	0.2–0.4		27 (100%)	0 (0.0%)	27 (100%)	0	
70–79	6 (75.0%)	2 (25.0%)	8 (100%)	0.24–0.26		8 (100%)	0 (0.0%)	8 (100%)	0		8 (100%)	0 (0.0%)	8 (100%)	0	
80+	5 (83.3%)	1 (16.7%)	6 (100%)	0.15–0.17		6 (100%)	0 (0.0%)	6 (100%)	0		5 (83.3%)	1 (16.7%)	6 (100%)	0.15–0.17	
Grand Total (N)	704 (80.8%)	167 (19.2%)	871 (100%)	0.18–0.20		538 (61.8%)	333 (38.2%)	871 (100%)	0.37–0.40		817 (93.8%)	54 (6.2%)	871 (100%)	0.5–0.7	

**Table 4 tropicalmed-08-00171-t004:** Cocirculation antibody seropositivity of ZIKV, FLAVI and malaria.

Region	Zika and Flavivirus Cocirculation						Flavivirus-Malaria Cocirculation						Zika-Malaria Cocirculation					
	Negative	Positive	Total Examined (N)	95% CI	OR	*p-Value*	Negative	Positive	Total Examined (N)	95% CI	OR	*p-Value*	Negative	Positive	Total Examined (N)	95% CI	OR	*p-Value*
South	141 (92.8%)	11 (7.2%)	152 (100%)		1	0.14	146 (96.1%)	6 (3.9%)	152 (100%)		1	0.24	141 (92.8%)	11 (7.2%)	152 (100%)		1	0.14
North	294 (98.0%)	6 (2.0%)	300 (100%)	0.70–11.4	2.8		297 (99.0%)	3 (1.0%)	300 (100%)	0.12–1.71	0.3		294 (98.0%)	6 (2.0%)	300 (100%)	0.70–11.4	2.8	
Central	416 (99.3%)	3 (0.7%)	419 (100%)	0.08–1.42	0.3		410 (97.9%)	9 (2.1%)	419 (100%)	0.58–8.09	2.2		416 (99.3%)	3 (0.7%)	419 (100%)	0.08–1.42	0.3	
Sex																		
Male	243 (96.4%)	9 (3.6%)	252 (100%)		1	0.27	246 (97.6%)	6 (2.4%)	252 (100%)		1	0.01	243 (96.4%)	9 (3.6%)	252 (100%)		1	0.28
Female	605 (97.7%)	14 (2.3%)	619 (100%)	0.68–3.74	1.6		617 (99.7%)	2 (0.3%)	619 (100%)	1.50–37.5	7.5		605 (97.7%)	14 (2.3%)	619 (100%)	0.67–3.72	1.5	
Domicile																		
Urban	495 (97.6%)	12 (2.4%)	507 (100%)		1	0.00	499 (98.4%)	8 (1.6%)	507 (100%)		1	0.00	495 (97.6%)	12 (2.4%)	507 (100%)		1	0.00
Rural	254 (98.4%)	4 (1.6%)	258 (100%)	0.03–0.32	0.1		254 (98.4%)	4 (1.6%)	258 (100%)	0.03–0.39	0.1		254 (98.4%)	4 (1.6%)	258 (100%)	0.03–0.32	0.1	
Slum	92 (86.8%)	14 (13.2%)	106 (100%)	3.10–30.1	9.6		94 (88.7%)	12 (11.3%)	106 (100%)	2.55–25.7	8.1		92 (86.8%)	14 (13.2%)	106 (100%)	3.10–30.1	9.6	
Pregnancy status																		
Pregnant	217 (93.1%)	16 (6.9%)	233 (100%)		1	0.00	230 (98.7%)	3 (1.3%)	233 (100%)		1	0.49	217 (93.1%)	16 (6.9%)	233 (100%)		1	0.00
Nonpregnant	634 (99.4%)	4 (0.6%)	638 (100%)	3.86–35.3	11.6		633 (99.2%)	5 (0.8%)	638 (100%)	0.39–6.96	1.7		634 (99.4%)	4 (0.6%)	638 (100%)	3.86–35.3	11.6	
HIV status																		
HIV positive	484 (97.0%)	15 (3.0%)	499 (100%)		1	0.19	492 (98.6%)	7 (1.4%)	499 (100%)		1	0.11	484 (97.0%)	15 (3.0%)	499 (100%)		1	0.19
HIV negative	366 (98.4%)	6 (1.6%)	372 (100%)	0.72–4.91	1.9		361 (97.0%)	11 (3.0%)	372 (100%)	0.17–1.21	0.5		366 (98.4%)	6 (1.6%)	372 (100%)	0.72–4.91	1.9	
Blood product source																		
Outpatient serum	756 (99.3%)	5 (0.7%)	761 (100%)		1	0.00	749 (98.4%)	12 (1.6%)	761 (100%)		1	0.36	756 (99.3%)	5 (0.7%)	761 (100%)		1	0.00
Blood bank serum	95 (86.4%)	15 (13.6%)	110 (100%)	0.01–0.11	0.4		110 (100%)	0 (0.0%)	110 (100%)	0.21–62.6	3.6		95 (86.4%)	15 (13.6%)	110 (100%)	0.01–0.11	0.0	
Grand Total (N)	718 (82.5%)	153 (17.5%)	871 (100%)	0.16–0.18			852 (97.8(%)	19 (2.2%)	871 (100%)				834 (95.7%)	37 (4.3%)	871 (100%)	0.3–0.5		

**Table 5 tropicalmed-08-00171-t005:** Age-specific seropositivity of Zika, malaria, and Zika-malaria.

Age (Years)	Zika-Flavivirus					Flavivirus-Malaria					Zika-Malaria					
	Negative	Positive	Total Examined (N)	95% CI	*p-Value*	Negative	Positive	Total Examined (N)	95% CI	*p-Value*		Negative	Positive	Total Examined (N)	95% CI	*p-Value*
0–9	3 (100%)	0 (0.0%)	3 (100%)	0	0.15	3 (100%)	0 (0.0%)	3 (100%)	0	0.02		3 (%)	1 (0.0%)	3 (100%)	0	0.05
10–19	51 (100%)	0 (0.0%)	51 (100%)	0		50 (98.0%)	1 (2.0%)	51 (100%)	0.1–0.3			46 (90.2%)	5 (9.8%)	51 (100%)	0.8–0.10	
20–29	195 (99.5%)	1 (0.5%)	196 (100%)	-0–0.1		193 (98.5%)	3 (1.5%)	196 (100%)	0.0–0.2			195 (96.4%)	7 (3.6%)	196 (100%)	0.2–0.4	
30–39	306 (96.5%)	11 (3.5%)	317 (100%)	0.2–0.4		312 (98.4%)	5 (1.6%)	317 (100%)	0.0–0.2			315 (96.8%)	10 (3.2%)	317 (100%)	0.2–0.4	
40–49	171 (94.5%)	10 (5.5%)	181 (100%)	0.4–0.6		178 (98.3%)	3 (1.7%)	181 (100%)	0.0–0.2			180 (95.0%)	9 (5.0%)	181 (100%)	0.4–0.6	
50–59	73 (89.0%)	9 (11.0%)	82 (100%)	0.10–0.12		80 (98.8%)	1 (1.2%)	82 (100%)	0.0–0.2			81 (95.1%)	4 (4.9%)	82 (100%)	0.3–0.5	
60–69	24 (88.9%)	3 (11.1%)	27 (100%)	0.10–0.12		26 (96.3%)	1 (3.7%)	27 (100%)	0.2–0.4			25 (%)	2 (7.4%)	27 (100%)	0.6–0.8	
70–79	6 (75.0%)	2 (25.0%)	8 (100%)	0.24–0.26		8 (100%)	0 (0.0%)	8 (100%)	0			0 (%)	0 (0.0%)	8 (100%)	0	
80+	5 (83.3%)	1 (16.7%)	6 (100%)	0.15–0.17		6 (100%)	0 (0.0%)	6 (100%)	0			0 (%)	0 (0.0%)	6 (100%)	0	
Grand Total (N)	834 (95.8%)	37 (4.2%)	871 (100%)	0.2–0.5		857 (98.4%)	14 (1.6%)	871 (100%)	0.0–0.2			833 (95.6%)	38 (4.4%)	871 (100%)	0.3–0.5	

## Data Availability

All data are contained in the manuscript.

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
