# Peer review of "Zika, Flavivirus and Malaria Antibody Cocirculation in Nigeria"

_tropicalmed, 2023, doi:10.3390/tropicalmed8030171_

Round 1
Reviewer 1 Report
Is malaria a viral disease? if not why was it lumped with the viral disease as if it is? Malaria was mentioned in 105 places in the write-up including references, with no recourse to the cause and to clearly separate it from other viral diseases transmitted by the same vector. The impression of a lay reader would be that a virus is the cause of malaria. This should be clarified

Author Response
REVIEW 1
Is malaria a viral disease? if not why was it lumped with the viral disease as if it is? Malaria was mentioned in 105 places in the write-up including references, with no recourse to the cause and to clearly separate it from other viral diseases transmitted by the same vector. The impression of a lay reader would be that a virus is the cause of malaria. This should be clarified.
ANSWER: Malaria is not a viral disease, it’s a parasitic infection.
Vector-borne diseases such as Malaria, FLAVI, and ZIKV have similar endemic profiles and symptoms, with potentially fatal outcomes if left undetected. It is crucial to comprehend the prevalence and distribution of these diseases to enhance diagnosis and develop therapeutic interventions. These infections have recently been a worldwide concern, particularly in tropical and subtropical areas like Nigeria, due to recurring outbreaks. Several of these diseases have become endemic to regions where malaria is widespread, causing millions of occurrences each year. Malaria is a significant public health threat, is caused and spread by five different species of protozoal parasite, Plasmodium. These include P. falciparum, P. ovale, P. malariae, P. vivax and P. knowlesi that are carried and spread by Anopheles mosquito [4, 5].
Malaria, ZIKV, and FLAVI have a similar epidemic pattern that mostly affects tropical regions worldwide. Several studies have shown that all three diseases can co-circulate [10, 11]. The diseases share comparable clinical manifestations where fever is the most common symptom. The burden of these infections has surged due to frequent outbreaks in different regions of Nigeria. Factors such as global travel and rapid urbanization have introduced vector populations to new environments, contributing to the expansion of disease endemicity [14]. Consequently, the co-circulation of Malaria with FLAVI and ZIKV complicates the diagnosis and treatment process.

Reviewer 2 Report
The manuscript entitled "Zika, Flavivirus and Malaria Antibody Cocirculation in Nigeria." Title, abstract and overall rationale of work is well written. However, there are still some major concerns, which needs to be addressed before publication.
1) In abstract part: Introduction section is not written well author need to revised and I suggest to author remove the reference in the abstract part.
2) Background section: No need to repeat same sentences that is already mention in the abstract part and author need to remove or revise.
3) Material methods section: In Laboratory testing section author need to elaborate and write details how they perform experiments.
4) Figure 1 resolution is not good especially written part, author should revise.
4) Results section: Table 2, 3, 4 and 5 are not clear and author must be present well manner. Furthermore, the figures quality are low and author should increase resolution of all these figure.
However, Results section written well and describe properly.
5) Discussion section: Much more explanations and interpretations must be added for the results, which are not enough at all. It is suggested to compare the results of the present research with some similar studies which is done before. Author must be write concise way.
6) Conclusion section must be elaborate and this section should present at least in one 250-300 words paragraph and author must write future prospective and significance of this study.
7) There are some of punctuation and typographical errors throughout in the manuscript. kindly correct it
Author Response
REVIEWER 2
1) In abstract part: Introduction section is not written well author need to revised and I suggest to author remove the reference in the abstract part.
ANSWER: Arboviruses and malaria are a significant cause of mortality worldwide. Their widespread diffusion, cocirculation, and lack of broad-spectrum antivirals, vaccines, and the resistance to antimalaria present a global public health challenge. The co-circulation of Zika virus, flaviviruses and malaria infection is of particular concern in numerous countries in the tropics and subtropics. These mosquito-borne infections have clinical presentations that overlap with other diseases (dengue, West Nile virus, and Japanese encephalitis, chikungunya onyong onyong,), making it a diagnostic challenge for clinicians in regions where they co-circulate. In urban settings, where these diseases are endemic and share common biological, ecological, and economic factors, they may impact treatment outcomes and lead to epidemiological synergy
2) Background section: No need to repeat same sentences that is already mention in the abstract part and author need to remove or revise.
ANSWER. Reworked as suggested.
3) Material methods section: In Laboratory testing section author need to elaborate and write details how they perform experiments.
ANSWER: For malaria: In brief, 5 µL of blood sample was transferred into the sample well using the appropriate device included in the kit, and five drops of lysis buffer were added to the buffer well. The results were read visually after 15-20 minutes.
For ZIKV and FLAVI. . In brief, test strips were loaded with ZIKV and FLAVI antigens and incubated with diluted serum in a dish for 1 hour. The strips were then washed three times. Peroxidase-conjugated anti-human antibodies (IgG-specific) were added, incubated for 45 min, and washed three times. Insoluble bands developed at the sites on the test strips occupied by antibodies 8 min after the addition of the colouring solution.
4) Figure 1 resolution is not good especially written part, author should revise.
ANSWER: Revised as instructed
4) Results section: Table 2, 3, 4 and 5 are not clear and author must be present well manner. Furthermore, the figures quality are low and author should increase resolution of all these figure.
ANSWER: Revised and resolution increased. Also, the tables are better viewed in web out. we advised that you view the tales in a web out please.
However, Results section written well and describe properly.
5) Discussion section: Much more explanations and interpretations must be added for the results, which are not enough at all. It is suggested to compare the results of the present research with some similar studies which is done before. Author must be write concise way.
ANSWER: We have made some additions as suggested by the reviewer with respect to what we have investiaged and at this point feel/felt we have exhausted and related our findings well enough to consummate what is required. Many thanks.
6) Conclusion section must be elaborate and this section should present at least in one 250-300 words paragraph and author must write future prospective and significance of this study.
ANSWER: We have also made some additions to cover what investigated and put forward some recommendations.
7) There are some of punctuation and typographical errors throughout in the manuscript. kindly correct it.
ANSWER: fixed.
Many thanks for taking your precious time to review our paper.
Kind regards,
Peter on behalf of co-authors.

Round 2
Reviewer 2 Report
The authors have addressed all the concerns raised in the previous version of the manuscript and the quality has much improved after incorporating required modifications. Therefore, the manuscript may be considered for publication in this Journal.